# Hormonal (Im)Balance and Reproductive System’s Disorders in Transplant Recipients—A Review

**DOI:** 10.3390/biology10040271

**Published:** 2021-03-26

**Authors:** Dagmara Szypulska-Koziarska, Kamila Misiakiewicz-Has, Barbara Wiszniewska

**Affiliations:** Department of Histology and Embryology, Pomeranian Medical University in Szczecin, Powstańców Wielkopolskich 72, 70-111 Szczecin, Poland; kamila.misiakiewicz@pum.edu.pl (K.M.-H.); barbara.wiszniewska@pum.edu.pl (B.W.)

**Keywords:** transplantation, immunosuppression, kidney, liver, heart, bone marrow, tacrolimus, cyclosporine A, prednisone, azathioprine

## Abstract

**Simple Summary:**

Nowadays, the average human life expectancy is increasing. This applies to both healthy and chronically ill people. It is possible due to improvements in technology, living conditions, and better, more accessible medical care. As the number of patients with end-stage organ failure increases and due to great progress in modern transplantology, every year the number of transplantations rises worldwide. Organ transplantation is not only an ultimate form of therapy but also, especially nowadays, a life-saving procedure. Patients who have undergone transplantation need to face the problem of long-term immunosuppressive therapy on a daily basis, which prolongs the proper function of the grafted organ and prevents the development of graft-versus-host disease. On the other hand, numerous side effects are associated with the usage of these medicaments, among these are disturbances in sex-related hormones, therefore influencing fertility.

**Abstract:**

The rising need for treatment of end stage of organ failure results in an increased number of graft recipients yearly. The most commonly transplanted organs are kidney, heart, liver, bone marrow, lung and skin. The procedure of transplantation saves and prolongs the lives of chronically ill patients or at least improves the quality. However, following transplantation recipients must take immunosuppressive drugs on a daily basis. Usually, the immunosuppressive therapy comprises two or three drugs from different groups, as the mechanism of their action varies. Although the benefits of intake of immunosuppressants is undeniable, numerous side effects are associated with them. To different extents, they are neurotoxic, nephrotoxic and may influence the function of the reproductive system. Nowadays, when infertility is an urgent problem even among healthy pairs, transplant recipients face the problem of disturbance in the hypothalamic−pituitary axis. This review will provide an overview of the most common disturbances among the concentration of sex-related hormones in recipients of both sexes at different ages, including sexually immature children, adults of reproductive age as well as elderly women and men. We have also focused on the numerous side effects of immunosuppressive therapy regarding function and morphology of reproductive organs both in males and females. The current review also presents the regimen of immunosuppressive therapy and time since transplantation.

## 1. Introduction

Due to the great progress in immunology, pharmacology and surgery in the last decades of the 20th century, a rising number of patients can benefit at present from the wider and better results of modern transplantology. Each year, the number of graft recipients increases worldwide. For obvious reasons, this results from increasing demand, however it has become possible due to the simultaneous development of modern, cutting-edge techniques in surgery and the rising quality of postoperative treatment as well as the higher effectiveness of immunosuppressive drugs [1]. The majority of all organ transplantations, such as heart, lungs, skin, bone marrow or liver, save or at least prolong the lives of patients. Concomitantly, successfully carried out procedures improve the quality of life in the case of renal transplant recipients. Nevertheless, following transplantation, with the exception of autologous transplants, patients are bound to face the problems connected with immunosuppressive therapy for the rest of their lives, as the immune system of recipients engages in a range of mechanisms aimed at overcoming foreign agents, which can contribute to graft failure. The rejection of allografts is mediated primarily by infiltration of the graft by T lymphocytes and the range of inflammatory reactions that arise due to their presence. The proliferation of mononuclear leukocytes, such as lymphocytes and monocytes, is a clear demonstration of acute and chronic rejection within the allograft [2]. Modern immunosuppressive agents constitute four groups of drugs. These are (i) calcineurin inhibitors including cyclosporin A and tacrolimus; (ii) mTOR inhibitors including sirolimus and its analog everolimus; (iii) inhibitors of the inosine monophosphate dehydrogenase consisting of mycophenolate mofetil, mycophenolate sodium and azathioprine; (iv) glucocorticoids, mostly prednisone [3,4]. Immunosuppressive drugs are widely applied to avoid graft rejection. However, they are known to exert deteriorating effects on the organism, affecting, among others, the endocrine system, including the reproductive system [5,6]. Nowadays, when infertility is a rising problem in a healthy human population, it is estimated that every fifth couple is already infertile [7].

This paper focuses on the description and summary of only the latest scientific reports of the past 20 years regarding the sex hormone profile and disorders associated with the reproductive system in transplant recipients undergoing different posttransplant conditioning. Taking into account the availability of literature, the most frequently transplanted organs in last 20 years are as follows: kidney, liver, bone marrow and hematopoietic stem cells (HSC), heart.

## 2. Renal Transplant Recipients

The final stage of kidney failure frequently leads to moderate or severe sexual dysfunction [8]. The imbalance of sex hormones is frequently diagnosed in dialyzed patients and mostly a decrease in the concentrations of T, DHT, DHEA, DHEAS are noticed, as well as elevated levels of FSH, LH or PRL. According to the latest data, approximately 50–70% of male and female patients with chronic kidney disease (CKD) suffer from sexual dysfunction [8]. Renal transplantation is one of the kidney replacement treatments. In contrast to transplantation of other solid organs, renal transplantation is not a life-saving treatment, but the one which successfully prolongs life expectancy and improves the quality of life, thanks to which, patients may return to their professional and social roles, including sexual activity [1]. Chronic kidney disease results in dysfunction of the hypothalamic-pituitary-gonadal axis both in males and females. The severity of the endocrine and reproductive dysfunction seems to be proportional to the degree of kidney impairment [9]. Renal transplant recipients are patients who most frequently suffered from uremia for many years before transplantation. The latter is the highly deteriorating state of the organism, mostly affecting reproductive function, as only 10% of dialyzed patients can get pregnant. Among the patients with CKD, the impairment of the hypothalamic-pituitary-gonadal axis is frequently diagnosed [7,10]. The disturbances in the hypothalamic-pituitary-gonadal axis are manifested differently, depending on sex. Almost 70% of CKD women suffer from disrupted ovarian cycles, shortened luteal phase, anovulatory cycles or premature menopause for approximately 4.5 years [11]. Among men, disorders in the hypothalamic-pituitary-gonadal axis are usually reflected in erectile dysfunction, lowered libido, improper spermato- and steroidogenesis, most of which result from hypotestosteronemia [12]. It is believed that a lowered concentration of circulating sex steroidal hormones does not only result in clinical hypogonadism but may be additionally associated with a higher risk of premature cardiovascular disease development and graft loss as it plays a significant role in pathogenesis and the progression of kidney failure [9,13,14].

According to the latest data, the physiological levels of sex hormones are restored approximately six months after successful kidney transplantation, as uremia is not observed then anymore [1,12,15]. On the other hand, the recipients need to undergo immunosuppressive therapy daily, which may influence restored hormonal homeostasis. The latest literature available concerning renal transplantation and the sex hormone profile is based on two main trends; the disturbances of the hypothalamic-pituitary-gonadal axis in dialyzed/CKD patients versus grafted patients and the comparison of the sex hormone profile in recipients depending on the immunosuppressive therapy used.

### Renal Transplant Recipients vs. CKD/Dialyzed Patients

When discussing the influence of kidney transplantation in the spectrum of hormonal balance, attention should be paid to the period of time following transplantation. As mentioned above, the hormonal balance should be restored approximately 5–6 months after successful transplantation. On the other hand, the influence of immunosuppressive therapy should not be omitted. The literature available provides information from different periods following transplantation, therefore all the data may be divided into the short- or long run categories.

In the study of Teng [16] the group of grafted men of reproductive age, undergoing immunosuppressive therapy including cyclosporin A + azathioprine + prednisone, was analyzed for the occurrence of sex hormone profile imbalance and the quality of sperm. Nevertheless, 1–2 months from transplantation, in all men the concentration of T was comparable with pre-transplantation results, and it was lowered below control. Not surprisingly the levels of PRL and LH were elevated when compared with the healthy control, however lowered in comparison to the pre-transplantation result. Such a short period of time was probably not enough to restore the homeostasis, but only 3–4 months after transplantation, the concentrations of T, LH, FSH and PRL were comparable with the results of the healthy control. Interestingly, semen parameters, such as volume or density, were lowered when compared with control, however the sperm motility and sperm survival rate as well as sperm normal morphology were within a normal range [16].

The disorders regarding the bone structure and metabolism parameters were also observed in grafted men whose immunosuppressive therapy included cyclosporin A, azathioprine and prednisone. Although, the normal concentration of T in blood was maintained, the decreased volume and wall thickness, reduced number of osteoblast and osteoclast, decreased osteoid and mineralizing surface as well as reduced appositional rate were frequently diagnosed [17,18]. Perhaps in the long-term therapy, male recipients are able to compensate for the secretion of T, and therefore it may be concluded that the level of testosterone is not the factor affecting bone status. In this regard, these non-sex hormone factors may be in example: time after transplantation, old age, function of kidney, length of uremia or immunosuppressive drugs.

There are studies which evaluate the long-term influence of immunosuppressive therapy including cyclosporin A, tacrolimus, azathioprine and mycophenolate mofetil, in kidney-grafted men of various ages, ranging from young, middle-age to older men [19,20]. A disrupted hypothalamic-pituitary-gonadal axis was noted in a group of these patients. It was manifested by impairment in sex hormone profile and anomalies concerning the male reproductive system. Among others, significantly elevated PRL concentration as well as lowered below the norm were concentrations of A, T, DHEAS, and it was reflected in decreased testicular and prostate volume together with reduced libido and erectile dysfunction [19,20,21,22].

Although adequate recovery of sexual/reproductive function is usually revealed approximately six months after transplantation, it is not infrequent for hypothalamic-pituitary-gonadal axis inhibition to persist. Perhaps, it depends on the regimen of immunosuppressive drugs that is administered to patients following transplantation. It is not surprising that immunosuppressive treatment influences steroidogenic pathway. What is interesting, glucocorticoids are known to suppress GnRH release, and it was mirrored by the research of Tauchmanova et al. and Teng et al. [19,20] where male patients did not suffer from affected GnRH secretion, but testicular release instead. It seems probable that the long-term usage of immunosuppressive drugs affects the testicular tissue, but does not influence the hypophysis, indicating that immunosuppressive drugs, especially prednisone, may be toxic for the reproductive system. Patients who are given prednisone should be looked after by specialists, as lowered T level is associated with hypertension, insulin resistance, abdominal obesity, an elevated level of inflammation markers or increased rate of cardio-vascular diseases [15]. There are studies on men comparing hormonal equilibrium in the same group of patients before and after transplantation. Hereby, it was noticed that the concentration of E2 and PRL was reduced [16], whereas the level of T was elevated [16,22,23,24,25] in comparison to patients before transplantation. What is interesting, although in recipients the level of T was improved, men suffered from erectile dysfunction with differing frequency [16,22,25]. The etiology of erectile dysfunction is multifactorial, and testosterone deficiency plays a great role, although not the only one. It is still unclear why in the case of some patients with a normal concentration of T, the erectile dysfunction is diagnosed, whereas in some patients with T deficiency the erectile function is preserved. Most probably, it can therefore be concluded, that erectile function does not directly depend on the T level. The first scientific studies reporting erectile dysfunction in uremic patients came to light in 1975 and, although it was 45 years ago, this problem is still current. Little is known, according to the most up-to-date literature about the influence of particular immunosuppressive drugs in comparison to others. Usually, research does not pay enough attention to the comparison between two or more immunosuppressive protocols, instead, they focus on the comparison of the hormone levels before and after transplantation.

Sirolimus is nowadays believed to be one of the most harmful immunosuppressive drugs to the reproductive system. In terms of this fact, the latest reports on the influence of sirolimus on sex-related hormones in kidney-grafted men seems to be confusing. In transplanted men, inhibition of T secretion was observed frequently under such immunosuppressive therapy [13,25,26]. Some clinicians noted an increased level of FSH and LH [26], whereas others observed the levels of FSH and LH within the regular range [13,15], yet Cavanaugh et al. [27] revealed reduced concentration of FSH and LH. The central hypogonadotropic releasing mechanisms may be influenced by an intake of sirolimus in various combinations, but on the other hand, some data indicate proper physiologic release. More probably, it seems that the inhibition of T secretion could be at the receptor or even post receptor level, among others affecting cholesterol transformation [26]. Simultaneously, only in patients in which the concentration of FSH and LH were within the normal range, in overweight, hypertriglyceridemia, hypertension or diabetes were seen [28], whereas the other patients did not report any complaints. Nevertheless, patients of reproductive age who have undergone immunosuppressive therapy including sirolimus should be monitored by the clinicians.

In modern immunosuppressive treatment, immunoglobulins, for instance basiliximab, are used by renal transplant recipients quite rarely. However, among the men of reproductive age on such a therapy, six months after surgery, the proper secretion of T was restored. On the other hand, the concentration of FSH and LH increased, suggesting an impairment in the hypothalamic-pituitary-gonadal axis. Moreover, grafted patients complained about infections, anemia, posttransplant diabetes, cholesterolaemia, triglyceridemia and hypertension, of various intensity levels and occurring with different frequency [17]. Comparable research was performed on grafted women and the same trend of hypothalamic-pituitary-gonadal axis disturbances were visible [19]. Inversed results were observed by Kim et al. [18], which might be related to a different immunosuppressive regimen that was applied. Surprisingly, approximately five years after transplantation, the concentration of E2 in grafted women of reproductive age was not established to its level in the healthy control. In fact, it was significantly elevated. Moreover, the concentration of FSH was elevated and the levels of progesterone and LH were lowered vs. control [19]. Perhaps different drug regimens influenced the endocrine system in a different manner. Therefore, more research was taken into account in this paper. Thus, Kim et al. [18] compared the influence of prednisone on hormonal balance in female recipients of reproductive age, however no such influence was noticed. What is interesting, although no differences in sex hormone profile were detected irrespective of whether the patients were given prednisone or not, there were some differences noted in the occurrence and severity of reproductive system disorders. In renal transplant recipients whose immunosuppressive therapy excluded prednisone, amenorrhea, uterine bleeding, infertility, irregular menstruation, dysmenorrhea, menometrorrhagia were observed with greater intensity when compared to recipients receiving prednisone [18]. Although most disturbances were milder in the prednisone-receiving patients, the infertility was much more pronounced in this particular group. It can, therefore, be concluded that glucocorticoids, likewise prednisone, in the long run may highly contribute to infertility in woman after transplantation.

In the long term, immunosuppressive therapy in grafted females, especially of reproductive age, most often a multidrug regimen including cyclosporin A/tacrolimus, azathioprine/mycophenolate mofetil and prednisone is applied, as these immunosuppressive drugs are believed to be relatively safe for embryo and fetus in case of pregnancy. However, the literature reveals a significant decrease in the concentration of E2, as well as an increased level of FSH in the blood of such patients [1,17]. Additionally, recipients suffered from numerous disorders, including reduced endometrial thickness, irregular menstrual bleeding and decreased parameters of bone structure parameters, i.e., volume, wall thickness, osteoblasts number and function, osteoclasts number, osteoid surface, mineralizing surface, as well as appositional rate [1,17]. The aforementioned results suggest that women of reproductive age with a stable, long-term functioning kidney graft may develop hypothalamic-pituitary-gonadal axis disorders that would be manifested in lowered E2 concentration. However, the level of E2 seems to be dependent on the dose of cyclosporin A or glucocorticoid administered [17]. It is not surprising that estrogen deficiency may widely affect the quality of life and health of women. It is already known that estrogens play a great role in the structural integrity and metabolism of bones. The deficiency of E2 level results in the imbalance between bone resorption and formation, leading to osteoporosis. Therefore, attention should be paid to the potential pros and cons of sex hormone replacement therapy in female renal transplant recipients, especially considering long term immunosuppressive therapy.

There are some studies available which indicate that steroidogenesis in kidney-grafted women under immunosuppressive therapy may be strongly affected. In patients with stable graft function a few years after transplantation, the regimen may be changed in comparison to the one applied right after the transplantation. There are indications to reduce the dose and the number of immunosuppressive medicines included in the treatment protocol. Most often, patients with stable kidney function use calcineurin inhibitors as these immunosuppressive drugs are believed to be relatively safe for recipients. On the other hand, cyclosporin A and tacrolimus have been seen to interfere with estradiol binding to the estrogen receptor when their concentration is high [1]. Within a group of kidney transplanted women under calcineurin inhibitors therapy, hormonal imbalance with numerous disorders of the reproductive system were noticed [1]. Patients suffered, among others, from irregular menstruation, prolonged menstrual cycle and extremely reduced fertility manifested in anovulatory cycles. These alterations are most probably a consequence of hormonal imbalance manifested by an increased E2 level and decreased progesterone concentrations. Interestingly, the pituitary hormones LH, FSH and PRL were within the normal range. This may suggest that calcineurin inhibitors affect the steroidogenic pathway of cholesterol, which is a precursor for both E2 and progesterone, but at the same time calcineurin inhibitors do not inhibit nor accelerate synthesis of trophic hormones such as LH, FSH nor PRL. Keeping in mind that studied women were of reproductive age, these side effects of immunosuppressive therapy may greatly deteriorate the general health condition, well-being and social life of patients, since a chance for successful pregnancies seems to be greatly reduced. Moreover, a prolonged state of increased E2 level significantly raises the risk of cancer development [1]. There are other studies confirming the impact of different immunosuppressive regimens in women of reproductive age on hormonal equilibrium [1,17,19]. In all these patients prednisone was one out of two or one out of three immunosuppressive drugs, and the others were as follows: cyclosporin A or tacrolimus, azathioprine with cyclosporin A or mycophenolate mofetil with cyclosporin A. Although after the average of 3 years following transplantation, the level of FSH, LH, 17-OHP and androstenedione (A) were comparable with those of the control group, a significant decrease in the concentration of T and DHEAS was noted, when compared with the control women and concomitantly they were below the norm. Grafted women suffered not only from the hormone imbalance itself, but they also complained about the presence of ovarian cysts, decreased ovary volume, abnormalities with menstrual cycles and hirsutism. The authors believe that immunosuppressive therapy used daily may affect steroidogenic pathway and/or the ovarian tissue. This would explain why the hormonal imbalance was seen mostly in the case of steroidogenic hormones, but not in peptide ones.

Analogous research, as abovementioned was performed on postmenopausal women, and what was surprising, no hormonal imbalance was observed in this group of patients. Nevertheless, patients complained about the same sorts of disorders as premenopausal women with diagnosed hormonal imbalance [19]. It would not be pointless to consider a sex hormone replacement therapy in this group of patients as well as in younger patients.

There are a few studies in which the influence of particular immunosuppressive therapy is not considered, instead, the authors compare the hormone profile of grafted patients with the dialyzed patients with CKD. For instance, Grossmann [9] revealed that the levels of T and DHEA in a postmenopausal female approximately five years after transplantation, were lower in comparison to CKD and dialyzed patients. Concomitantly, the concentrations of DHT and E2 were comparable with CKD and dialyzed patients. Moreover, Grossmann et al. [12] observed a significant decrease in mortality of recipients vs. dialyzed patients, however in recipients vs. CKD the trend was opposite.

Most commonly noticed hormonal changes in male and female kidney-transplant recipients undergoing various regiment of immunosuppressive drugs described in the current chapter are presented in Figure 1. 

All the references cited in the current chapter are presented in the Table 1.

## 3. Liver Transplant Recipients

Nowadays, men constitute approximately 66% of all liver transplant recipients. It is important, since male patients suffering from end-stage of liver disease exhibit worse sexual function which is mirrored in hypogonadism and feminization. Among others, in these patients, testicular atrophy, decreased T level, reduced secondary sexual characteristics, muscle reduction, gynecomastia, decreased libido and fertility are frequently reported [24,30]. According to the latest reports, almost 50% of men with diagnosed cirrhosis, reduced spermatogenesis and peritubular fibrosis are noticed [33]. All the aforementioned symptoms may result in a regressed quality of professional, social and sexual life leading to high fatigue levels, depression or impotence [30]. Successful liver transplantation primarily saves the lives of patients suffering from end-stage liver disease but, secondly, it should improve the life quality of liver transplant recipients, among others, by restoring hormonal balance. Surprisingly, the conclusions of the latest literature are confusing. For instance, Chan et al. [34] did not observe any improvement in the sex hormone profile among male liver transplant recipients one or three months after successful transplantation. The concentrations of the following hormones: P, PRL, T, free T were comparable with those prior to the surgery. The only exception was the concentration of E2, which significantly decreased when compared with its level prior to transplantation. Concomitantly, all the men suffered from erectile dysfunction. Moreover, in the same group of men three months later, a reduced level of PRL turned out to be the only visible improvement. At the same time, erectile dysfunction was observed in fewer patients. It may be therefore concluded that erectile function depends more predominantly on the concentration of PRL, in comparison to the levels of T or E2. It is worth mentioning that all the patients were under tacrolimus therapy, which is believed not to be a reprotoxic immunosuppressive agent [30]. Similarly, confusing results achieved by Chiang et al. [30] when observing adult men aged approximately 54 years of age after one and up to six months from liver transplantation. In all these men, free T level as well as T and SHBG were observed to be reduced below the norm. Moreover, in all the patients the erectile dysfunction was diagnosed. Perhaps, restoring proper hormonal profile and sexual function after liver transplantation exceeds the period of six months. A 14-month observation in male liver transplant recipients was made by Iaria et al. [35], however in the case of two men only. Both of them were administered prednisone, basiliximab and tacrolimus or cyclosporin A. Interestingly, it was noticed that immunosuppressive therapy based on cyclosporin A resulted in highly elevated PRL and it was mirrored in gynecomastia, whereas in the case of the patient treated with tacrolimus, PRL concentration was within the normal range. The case report concerned two men only and thus it was probably not a representative trial, however the conclusion to be drawn is that cyclosporin A seems to be the immunosuppressive drug that affects hormonal balance to a higher degree, when compared with tacrolimus [35]. Nevertheless, it should be mentioned that the mechanism of cyclosporin A-related gynecomastia is not clearly comprehended, however, it is known that reduction of a dose of cyclosporin A or conversion from cyclosporin A to Tc or discontinuation of cyclosporin A treatment, restores hormonal status to that of before-the-surgery and concomitantly reduces gynecomastia [35,36]. Although gynecomastia is usually related with hyperprolactinemia, it is already known that cyclosporin A strongly inhibits the process of 2-hydroxylation and 17-oxidation of E2, therefore increasing the synthesis of E2 [35].

The data concerning the sex hormone profile in liver grafted women are scarcely found. This may be related to the fact that women constitute only one third of all liver transplant recipients. Nevertheless, successful liver transplantation in women, restores menstrual function, among others, therefore improving life quality [37]. The observation of female liver transplant recipients aged 46–55 years old, was performed three months after surgery. Surprisingly, the sex hormone profile was not improved by transplantation, since the concentration of E2 was reduced in comparison to the control group and women before transplantation. The concentrations of the remaining sex-related hormones, likewise FSH, LH, PRL, P and T were at comparable levels with those before surgery. Concomitantly, almost 46% of patients suffered from secondary amenorrhea and 8% complained about irregular menstrual cycles. The aforementioned lack of improvement in health condition, despite liver transplantation, may be related to the short period since the surgery. Nevertheless, more observations should be made to elucidate the influence of liver transplantation on sexual health in liver transplant women recipients.

Most commonly noticed hormonal changes in male and female liver-transplant recipients undergoing different regiment of immunosuppressive drugs described in the current chapter are presented in Figure 1. 

All the references cited in the current chapter are presented in the Table 2.

## 4. Bone Marrow Recipients

For over twenty years a great improvement in the treatment of numerous hematologic malignancies and aplastic anemia has been observed in modern hematology. This new mode of treatment is bone marrow transplantation. Since life expectancy of patients after transplantation is longer, attention has been paid increasingly to the quality of their lives in the long run. Bone marrow recipients frequently suffer from endocrine disturbances and these disorders may reflect in decreased fertility [38]. Gonadal damage shortly after transplantation is a very common abnormality, however it is known that sexually immature females may recover proper function of ovaries [39]. Nevertheless, each pregnancy after bone marrow transplantation is burdened with high risk and although there are some successful outcomes, spontaneous abortion as well as premature delivery and low weight of infants are the predominant rule [40]. Bone marrow transplantation may have an adverse effect on sexual life of females, despite hormone replacement [38] therapy. Moreover, sexual function may be also affected by the appearance of graft-versus-host disease [38]. Hovi et al. [38] enrolled into the research young females aged 16–33 years old who had undergone bone marrow transplantation with an average age of 12 years. Although the posttransplant treatment of all the patients differed, most of them were administered glucocorticoids, moreover, none of them underwent irradiation to the abdominal area. In all these patients, numerous disorders in hormonal status were observed, when compared to the norms. There were among others: an elevated FSH level as well as lowered concentration of T, A and DHEAS. Moreover, transplanted females suffered from ovarian dysfunction, irregular menstrual bleeding and lack of axillary and pubic hair [38]. The authors of this research have also noticed that 12 out of 15 patients who underwent transplantation prior to menarche, faced the problem of normal pubertal development. It was manifested by the lack of spontaneous menarche. There are some data available indicating that bone marrow transplantation in sexually immature girls may result in early menopause. Another group of young females aged 21–45 years old following hematopoietic stem cells transplantation was evaluated by Tauchmanova et al. [19]. Although the drugs protocols after transplantation differed among the patients, all of them were administered cyclosporin A with prednisone for approximately 1.5 years. The hormonal imbalance and numerous disorders of the female reproductive system were the problems that patients had to face. Females following transplantation, complained about hormonal imbalance reflected in the elevated FSH and LH levels, and decreased E2, T, A and DHEAS concentrations. The abovementioned abnormalities were mirrored in decreased ovarian volume and a reduced number of follicles per ovary, decreased uterine volume and thickness of the wall. As seen from this observation, recovery of normal reproductive function after bone marrow transplantation is not an early event, but rather a long-time process, that takes more than 1.5 years. It may be crucial information for couples in late reproductive age, since they could use new techniques of assisted reproduction, such as cryopreservation of embryos or usage of a surrogate uterus, some techniques which are nowadays on high quality level.

Most commonly noticed hormonal changes in male and female bone-marrow-transplant recipients undergoing various regiment of immunosuppressive drugs described in the current chapter are presented in Figure 1. 

All the references cited in the current chapter are presented in the Table 3.

## 5. Cardiac Transplant Recipients

In the latest literature available, still little is known about the sex-related steroids in patients after cardiac transplantation. The study of Fleischer et al. [41] seems to be valuable as it involved over a hundred men following cardiac transplantation over an average of three years. All the patients were administered prednisone and calcineurin inhibitors, predominantly cyclosporin A. From this study it is clearly visible that T levels were slightly below the norm in the 4th month after transplantation, but then gradually rose to achieve the normal level in the 6th and 24th month following surgery. Interestingly, the concentration of free T sustained the level greatly decreased below the norm for the whole duration of the study, although it also gradually rose over time. According to the concentration of E2, it rose as more time passed since transplantation, nevertheless the levels of E2 were within the normal limits for men. The hormone profile of non-steroidal sex-related hormones likewise LH and FSH remained within norms, although they were at their nadir initially, and rose gradually. Lowered T soon after grafting seems to be due to the central suppression of the hypothalamic-pituitary-gonadal axis, caused by a low level of LH that stimulates Leydig cells to produce androgens. Of note, although initially the concentration of T was lowered below the norm, patients did not complain of decreased libido, erectile dysfunction nor any other symptoms associated with the reproductive system. Seemingly, cyclosporin A with prednisone, used in the long run at precisely controlled doses, does not exert any deteriorating effect on gonadal function, which potentially makes it the well-suited protocol for these and other patients. This may be an important indication for clinicians, who select the regimen of immunosuppressive agents. Although it is known that the high doses of prednisone that are used initially after grafting suppress greatly the hypothalamic-pituitary-gonadal axis, as the surgery does, it is resolved by six months after transplantation, indicating recovery of hypothalamo−pituitary regulation [41]. A study that provides some insight into whether sirolimus affects male gonadal function was carried out by Kaczmarek et al. [42]. The scientist analyzed over 130 men following CT in the long term. When one group was administered sirolimus with mycophenolate mofetil or with tacrolimus, the other used protocols excluding sirolimus, likewise tacrolimus + mycophenolate mofetil, cyclosporin A + mycophenolate mofetil, tacrolimus + azathioprine or cyclosporin A + azathioprine. After approximately three years following transplantation it was obvious that patients on sirolimus regimens reported a reduced level of T in comparison to the group excluding sirolimus. Concomitantly sirolimus users had increased FSH and LH levels, which indicated a suppression of the hypothalamic-pituitary-gonadal axis. Moreover, the scientist observed decreased T/LH ratio in men using sirolimus, when compared with those on other immunosuppressants. As T deficiency results in upregulation of LH and FSH, a reduced T/LH ratio may be an indicator for testicular insufficiency [42]. Some research performed on rats showed that sirolimus usage leads to tubular atrophy and decreased testicular weight and sperm counts [43]. However, it is not surprising that sirolimus is an agent of high significance as far as acute allograft rejection is concerned. Its usage is associated with its valuable properties likewise antitumor and antifungal action. In the group of patients suffering now or in the past from tumor or persisting fungal diseases, sirolimus seems to be the most suitable immunosuppressive drug. On the other hand, sirolimus is strongly neurotoxic, causes arterial hypertension and may induce the development of post-transplant diabetes mellitus [44].

Most commonly noticed hormonal changes in male and female cardiac-transplant recipients undergoing various regiment of immunosuppressive drugs described in the current chapter are presented in Figure 1.

The comparison of the most commonly used regimens of immunosuppressive treatment used by the patients included in the current manuscript is presented in Figure 2. 

All the references cited in the current chapter are presented in the Table 4.

## 6. Summary

Without doubt, the impairment of hypothalamic-pituitary-gonadal axis in grafted patients is multifactorial. Taking into account the latest data, the proper function of both male and female gonads are usually improved approximately 5–6 months after successful transplantation. However, immunosuppressive therapy, administered daily, can adversely influence the homeostasis affecting restored gonad function. Therefore, it still remains unclear whether transplantation improves sexual and reproductive functions or not, especially in the long run. Nevertheless, in all patients after successful surgery, the sex hormone profile should be evaluated regularly, so that if any meaningful irregularities are revealed, appropriate replacement therapy can be implemented. It seems to be especially important for female patients of reproductive age, due to maternal readiness. Regardless of the percentage of transplanted females with preserved normal ovarian function, it seems to be of great importance to monitor them in order to select the most suitable contraception and family planning [38]. On the other hand, some disorders concerning the reproductive system, both male and female, may affect the health status and condition of the body leading to further abnormalities.

## 7. Conclusions

Concluding, clinicians need to consider all the advantages and disadvantages prior to the selection of a particular immunosuppressive drug to be applied in the protocol for grafted patients, especially in the long run, as side effects may accumulate and intensify over time.

## Figures and Tables

**Figure 1 biology-10-00271-f001:**
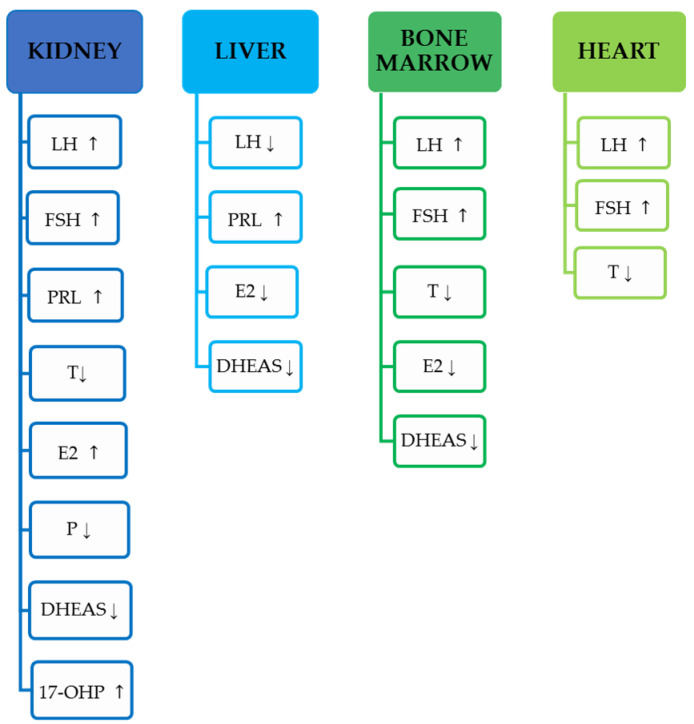
Most commonly noticed hormonal changes in men and women described in the current review, based on the transplanted organ. Increase in concentration indicated by ↑, a decrease in concentration indicated by ↓.

**Figure 2 biology-10-00271-f002:**
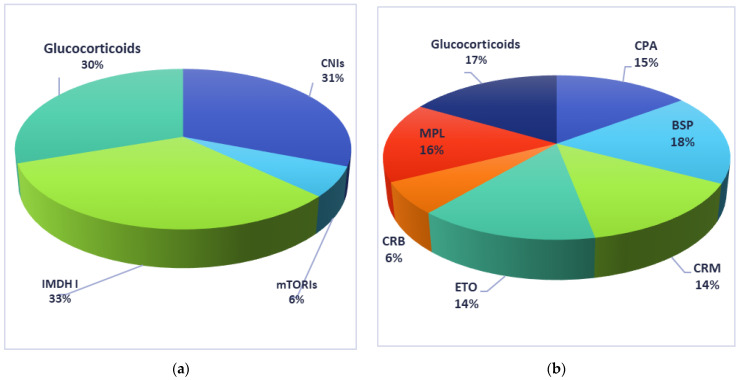
Immunosuppressive drugs used by recipients included in the current manuscript: (**a**) solid-organs recipients; (**b**) bone-marrow recipients. CNIs: calcineurin inhibitors; mTORIs: mTOR inhibitors; IMDH I: inhibitors for inosine monophosphate dehydrogenase; CPA: cyclophosphamide, BSP: busulphan; CRM: carmustine; ETO: etopside; CRB: cytarabine; MPL: melphalan.

**Table 1 biology-10-00271-t001:** Demographical and medical data of renal transplant recipients.

Conditioning (Number of Patients)	Sex (Number of Patients)	Age (Years)	Time after Tx	Hormonal Changes					Post-Tx Complications		Literature
CsA + AZA + PRE	M (735) + F (360)	25–34 (pregnant F)	mean 3.6 years	lack of information					gestational diabetes (28.6%); chronic transplant nephropathy during gestation and graft lost		[29]
CsA + AZA + PRE	M (68)	26–45	1–2 months	PRL: ↑ vs. C; ↓ vs. bTx					Semen volume: ↓ vs. C; ↑ vs. B		[16]
				LH: ↑ vs. C; ↓ vs. bTx							
				FSH: ↔ vs. C; ↔vs. bTx					Sperm motility: ↔ vs. C; ↑ vs. B		
				T: ↓ vs. C; ↔ vs. bTx							
			3–4 months	PRL: ↔ vs. C; ↓ vs. bTx					Sperm density: ↓ vs. C; ↑ vs. B		
				LH: ↔ vs. C; ↓ vs. bTx					Sperm survival rate: ↔ vs. C; ↑ vs. B		
				FSH: ↔ vs. C; ↔ vs. bTx					Sperm normal morphology: ↔ vs. C; ↑ vs. B		
				T: vs. C ↔; vs. bTx ↑							
CsA + AZA + PRE	F (32)	26–45	1–2 months	PRL: ↔ vs. C; ↓ vs. bTx					lack of information		[16]
				LH: ↑ vs. C; ↓ vs. bTx							
				FSH: ↑ vs. C; ↓ vs. B							
				E2: ↑ vs. C; ↑ vs. bTx							
CsA + AZA + PRE	F (32)	26–45	3–4 months	PRL: ↔ vs. C; ↓ vs. bTx					lack of information		[16]
				LH: ↔ vs. C; ↓ vs. bTx							
				FSH: ↔ vs. C; ↓ vs. bTx							
				E2: ↔ vs. C; ↑ vs. bTx							
CsA/Tac + MMF + PRE	F (31)	15–48	>3 months	E2: ↑ vs. C; P: ↓ vs. C LH: ↓ vs. C; ↑ FSH vs. C PRL: ↔ vs. C; TSH: ↔ vs. C					lack of information		[18]
A: CsA/Tac + MMF B: CsA/Tac + MMF + PRE	A: F (19) B: F (12)	15–48	A: mean 8 months B: mean 59.5 months	E2: vs. B ↔ P: vs. B ↔ LH: vs. B ↔ FSH: vs. B ↔ PRL: vs. B ↔ TSH: vs. B ↔					A	amenorrhea (10.5%); uterine bleeding (31.6%); infertility (5.3%); irregular menstruation (21.1%); dysmenorrhea (5.3%); menometrorrhagia (26.3%)	[18]
									B	amenorrhea (8.3%); uterine bleeding (8.3%); infertility (50%); irregular menstruation (8.3%); dysmenorrhea (8.3%); menometrorrhagia (16.7%)	
CsA + AZA + PRE	F (54)	31–52	mean 6.2 years	E2: 18.5 pg/mL FSH: 129.4 IU/L					menstrual bleeding: 1 per year; decreased endometrial thickness		[1]
lack of information	F (26)	50–62	4.9 years	T: ↓ vs. D & CKD DHT: ↔ vs. CKD & D E2: ↔ vs. CKD & D E1: ↔ vs. CKD & D DHEA: ↓ vs. D & CKD					Mortality: ↓ vs. D; ↑ vs. CKD		[12]
lack of information	M (44)	43–57	4.7 years	T: ↑ vs. D & CKD DHT: ↔ vs. D & CKD E2: ↓ vs. CKD; ↑ vs. D E1: ↑ vs. CKD; ↓ vs. D DHEA: ↔ vs. D & CKD T/E2 ratio: ↑ vs. D & CKD					BMI: ↑ vs. D; ↓ vs. CKD mortality: ↓ vs. D & CKD		[12]
CsA + PRE + AZA (3) AZA + PRE (5) CsA (8)	M (16)	40–54	120 months	T: 19.7 nmol/L PRL: 357.2 mU/L					↓parameters of bone structure: volume; wall thickness; osteoblast number and function; osteoid surface; osteoclast number; mineralizing surface; appositional rate		[19]
CsA + PRE + AZA (1) AZA + PRE (4) CsA (1)	pre-menopausal F (6)	37–43	142 months	E2: 209 pmol/L ^mean value of E2 level from follicular, phase, mid cycle and luteal phase^ PRL: 372.2 mU/L					↓ parameters of bone structure: volume; wall thickness; osteoblast number and function; osteoid surface; osteoclast number; mineralizing surface; appositional rate		[17]
CsA + PRE + AZA (1) AZA + PRE (3) CsA (4)	post-menopausal F (8)	53–59	123 months	E2: 93 pmol/L ^mean value of E2 level from follicular, phase, mid cycle and luteal phase^ PRL: 209.1 mU/L					↓ parameters of bone structure: volume; wall thickness; osteoblast number and function; osteoid surface; osteoclast number; mineralizing surface; appositional rate		[17]
PRE + CsA PRE + Tac PRE + AZA + CsA PRE + MMF + CsA	M (20)	23–44	14–75 months	FSH: ↔ vs. C; LH: ↔ vs. C; PRL: ↔ vs. C; 17-OHP: ↑ vs. C & elevated above norm; A: ↓ vs. C & lowered below norm T: ↓ vs. C & lowered below norm DHEAS: ↓ vs. C & lowered below norm					↓ prostate volume (100%); ↓ testicular volume (100%); ↓ libido (47%); erectile dysfunction (30%)		[20]
PRE + CsA PRE + Tac PRE + AZA + CsA PRE + MMF + CsA	F (20)	23–44	14–75 months	FSH: ↔ vs. C; LH: ↔ vs. C; PRL: ↑ vs. C; 17-OHP: ↔ vs. C; A: ↔ vs. C; T: ↓ vs. C ^lowered below norm^ DHEAS: ↓ vs. C ^lowered below norm^					ovarian cysts (15%); decreased ovarian volume (15%); abnormalities with menstrual cycles; hirsutism; POF (20%)		[20]
A: CsA + PRE (21) B: Tac + PRE (16)	M (37)	A: 38.7 B: 37.3	A: 73 months B: 46 months	FSH: A vs. B ↔ LH: A vs. B ↔ PRL: A vs. B ↔ T: A vs. B ↔					lack of information		[30]
A: CNIs (15) B: Rapa (15) C: Rapa + CNIs (29)	M (59)	48	mean 56 months	T: ↑ A vs. B; ↔ A vs. C; ↔ B vs. C					lack of information		[26]
CNIs (15) Rapa (15) Rapa + CNIs (29)	M (59)	48	mean 56 months	FSH: 13.7 mUI/mL ^elevated above norm^ LH: 14.7 mUI/mL ^within normal range^					lack of information		[26]
Rapa + PRE	M (15)	11–18	mean 81 months	T: ↓ over 2 years ^below normal range^ LH: ↓ over 2 years FSH: ↓ over 2 years					lack of information		[27]
EVE + CsA + PRE + Basiliximab	M (123)	<50	>6 months	T: 11.2 nmol/L ^within normal range;^ FSH: ↑ LH: ↑					infections (58.1%); anemia (29.05%); posttransplant diabetes (4%); increased cholesterol level (100%); increased triglycerides level (100%); hypertension (25.6%)		[28]
PRE/Flu + CsA PRE/Flu + AZA	A: M (38) B: ^premenopausal^ F (15) C: ^postmenopausal^ F (13)	23–70	A: 81 B: 89 C: 106	A B C	FSH: ^within normal range^ DHEAS: ^below normal range^				↓bone density; osteopenia (43%); osteoporosis (23%); hyperparathyroidism (100%)		[31]
				A C	LH: ^within normal range^ E2: ^elevated above norm^						
				LH	A: ^within normal range^ B: ^below normal range^ C: ^elevated above norm^						
				T	A; B: ^within normal range^ C: ^below normal range^						
Rapa + CNIs	M (32)	21–65	mean 22 months	free T: 11.6 ng/dL ^↔^ vs. ^C^ T: 393.3 ng/dL ^↓^ vs. ^C^ FSH: 12.8 mlU/mL ^↑^ vs. ^C^ LH: 10.9 mlU/mL ^↑^ vs. ^C^ PRL: 10.9 mlU/mL ^↔^ vs. ^C^					lack of information		[32]
CNIs	F (63)	18–44	mean 4.15 years	E2: 226.86 pg/mL ^increased^ vs. ^C^ FSH: 4.99 mlU/mL ^↔^ vs. ^C^ LH: 9.84 mlU/mL ^↔^ vs. ^C^ PRL: 18.64 ng/mL ^↔^ vs. ^C^ P: 15.05 ng/mL ^↓^ vs. ^C^					irregular menstruation (32%); prolonged menstrual cycle (31.9%); anovulatory cycles (55%)		[1]
lack of information	M (51)	>50 (51%) <50 (49%)	mean 7.2 years	free T: <40 pg/mL ^below normal range^ PRL: >15.5 ng/mL ^elevated above norm^					erectile dysfunction (65%)		[22]
CsA (19%) Tac (71%) Rapa (7%) EVE (3%) MMF/MMS (75%)	M (112)	54.6	mean 8.94 years	T: <350 ng/dL ^(in 52%) below normal range^ FSH: 5.9 Ul/L ^within normal range^ LH: 5.25 Ul/L ^within normal range^					overweight (52%); hypertriglyceridemia (100%); hypertension (37.5%); diabetes (26%)		[13]
CNIs + PRE + MMF	M (197)	A: 50–59 (44%) B: 60–69 (33%) C: ≥70 (23%)	17–35 months	T					↓ prostate volume		[20]
				A 10.3 nmol/L ^↓ vs C^		B 9.01 nmol/L ^↓ vs C^		C 7.67 nmol/L ^↓ vs C^			
lack of information	M (35)	lack of information	lack of information	T: 4.32 ng/mL ^↑^ vs. ^bTx^ E2: 19.57 pg/m ^↓^ vs. ^bTx^ PRL: 8.59 mIu/mL ^↓^ vs. ^bTx^					erectile dysfunction (54%)		[16]
lack of information	F (55)	18–40	1–5 years	E2: 205.9 pg/mL ^↑^ vs. ^C^ P: 13.2 ng/mL ^↓^ vs. ^C^ T: ^↓^ vs. ^C^ FSH, LH, PRL: ^↔^ vs. ^C^					irregular menstrual cycles (27.3%)		[1]
lack of information	M (25)	53.5	124 months	T: 515.7 ng/dL ^↑^ vs. ^bTx^					erectile dysfunction		[25]
A: PRE + CNIs + MMF B: PRE + Rapa/EVE + MMF	M (75)	A: 40.9 B: 41.2	>6 months	A			B		lack of information		[15]
				T: 8.8 nmol/L ^↑^ vs. ^bTx^ FSH: 7.7 mLU/mL ^↓^ vs. ^bTx^ LH: 6.3 mLU/mL ^↓^ vs. ^bTx^ PRL: 10.5 ng/mL ^↓^ vs. ^bTx^			T: 8.2 nmol/L ^↑^ vs. ^bTx^ FSH: 8.6 mLU/mL ^↓^ vs. ^bTx^ LH: 7.3 mLU/mL ^↓^ vs. ^bTx^ PRL: 11.2 ng/mL ^↓^ vs. ^bTx^				

M: male; F: female; CsA: cyclosporine A; AZA: azathioprine; PRE: prednisone; Tac: tacrolimus; MMF: mycophenolate mofetil; Rapa: rapamycin; CNIs: calcineurin inhibitors; Tx: transplantation; P: progesterone; C: control group; bTx: before Tx; D: patients under dialysis; CKD: patients with chronic kidney disease; ↓: decrease; ↑: increase; ↔: no statistical difference.

**Table 2 biology-10-00271-t002:** Demographical and medical data of liver transplant recipients.

Conditioning (Number of Patients)	Sex (Number of Patients)	Age (Years)	Time after Tx	Hormonal Changes	Post-Tx Complications	Literature
A Tac + PRE + T (9)	A M (6); F (3)	A 51	>6 months	A free T: 1.2 ng/dL ^↑^ vs. ^bTx^ T: 788 ng/dL ^↑^ vs. ^bTx^ E2: 45 ng/mL ^↑^ vs. ^bTx^ LH: 4.1 mIU/mL ^↓^ vs. ^bTx^	B ↓ albumin level (100%); deaths on average 84 days post-transplant (80%)	[38]
B Tac + PRE (5)	B M (3); F (2)	B 53		B T: 350 ng/dL ^↔^ vs. ^bTx^		
A CsA + Basiliximab + PRE	M (1)	55	A 6 months	A LH: 11.8 mU/mL ^below normal range^ T: 2.5 ng/mL ^below normal range^	A gynecomastia	[35]
B Tac + Basiliximab + PRE			B 15 months	B LH: 6.9 mU/mL ^within normal range^ T: 3.8 ng/mL ^within normal range^		
A CsA + Basiliximab + PRE	M (1)	64	A 13 months	A PRL: 778 mUI/L ^elevated above norm^	A gynecomastia	[35]
B Tac + Basiliximab + PRE			B 14 months	B PRL: 226 mUI/L ^within normal range^		
lack of information	M (41)	53.86	1 month 3 months 6 months	T: 3.11 pg/mL Free T: 16.75 pg/mL	erectile dysfunctions (100%); ↓ level of SHBG (100%)	[30]
lack of information	F (13)	46–55	3 months	E2: 49.12 pg/mL ^↓^ vs. ^C & bTx^ FSH: 38.25 mlU/mL ^↔^ vs. ^C & bTx^ LH: 26.12 mlU/mL ^↔^ vs. ^C & bTx^ PRL: 19.65 ng/mL ^↔^ vs. ^C & bTx^ P: 15.05 nmol/L ^↔^ vs. ^C & bTx^ T: 0.36 ng/mL ^↔^ vs. ^C & bTx^ DHEAS: 66.58 μg/dL ^↓^ vs. ^C;^ ^↑^ vs. ^bTx^	secondary amenorrhea (46%); irregular cycles (8%);	[37]
Tac	M (28)	55.3	6 months	P: ↔ vs. bTx; PRL: ↓ vs. bTx T: ↔ vs. bTx Free T: ↔ vs. bTx TSH: ↔ vs. bTx E2: ↓ vs. bTx	↓ free T vs. reference range	[34]
Tac	M (28)	55.3	1 month	P: ↔ vs. bTx PRL: ↔ vs. bTx T: ↔ vs. bTx Free T: ↔ vs. bTx TSH: ↑ vs. bTx E2: ↓ vs. bTx	↓ erectile dysfunction vs. bTx	[34]

M: male; F: female; CsA: cyclosporine A; AZA: azathioprine; PRE: prednisone; Tac: tacrolimus; MMF: mycophenolate mofetil; Rapa: rapamycin; CNIs: calcineurin inhibitors; Tx: transplantation; P: progesterone; C: control group; bTx: before Tx; D: patients under dialysis; CKD: patients with chronic kidney disease; ↓: decrease; ↑: increase; ↔: no statistical difference.

**Table 3 biology-10-00271-t003:** Demographical and medical data of bone marrow transplant recipients.

Conditioning (Number of Patients)	Sex (Number of Patients)	Age (Years)	Time after Tx	Hormonal Changes	Post-Tx Complications	Literature
BSP + CPA + CsA + PRE CRM + ETO + CRB + MPL + CsA + PRE	F (22)	21–45	12–24 months	FSH: ↑ vs. C; LH: ↑ vs. C; E2: ↓ vs. C; T: ↓ vs. C A: ↓ vs. C; DHEAS: ↓ vs. C PRL: vs. C ↔	↓ ovarian volume; ↓ uterine volume; ↓ number of follicles per ovary; ↓ endometrial thickness	[19]
BSP + CPA/CRM + ETO + CRB + MPL	F (23)	21–45	12–24 months	FSH: ↑ vs. C; LH: ↑ vs. C; E2: ↓ vs. C; T: ↔ vs. C; A: ↔ vs. C; DHEAS: ↔vs. C PRL: ↔ vs. C	↓ ovarian volume; ↓ uterine volume; ↓ number of follicles per ovary; ↓ endometrial thickness	[19]
BSP + CPA + PRE (31%) BSP + CRM + ETO + MPL + PRE (59%) BSP + MPL + PRE (11%)	A M (47)	17–55	3 months 12 months	A 3 months FSH: 22 U/L ^elevated above norm^ LH: 7.5 U/L ^normal range^ PRL: 6.9 ng/mL ^normal range^ T: 3 pg/mL ^normal range^ A 12 months FSH: 16.4 U/L ^elevated above norm^ LH: 6.2 U/L ^normal range^ PRL: 7 ng/mL ^normal range^ T: 4.8 pg/mL ^normal range^	A&B 3 months hyperthyroidism (16%); hypothyroidism (9%); “low T3 syndrome” (29%); antibodies to thyroid (12%); chronic thyroiditis (4.2%)	[19]
	B F (48)			B 3 months FSH: 65 U/L ^elevated above norm^ LH: 28 U/L ^elevated above norm^ PRL: 11 ng/mL ^normal range^ E2: 15 pg/mL ^below normal range^ B 12 months FSH: 49.5 U/L ^elevated above norm^ LH: 17 U/L ^elevated above norm^ PRL: 12.5 ng/mL ^normal range^ E2: 29.2 pg/mL ^below normal range^		
lack of information	F (24)	4–19	mean 9 years	FSH: ↑ above norm (95%) T: ↓ below norm (43%) A: ↓ below norm (63%) DHEAS: ↓ below norm (34%)	ovarian dysfunction; irregular menstrual bleeding; irregular menstruation; lack of axillary & pubic hair	[38]

M: male; F: female; PRE: prednisone; BSP: busulphan; CPA: cyclophosphamide; CRM: carmustine; ETO: etoposide; CRB: cytarabine; MPL: melphalan; Rapa: rapamycin; Flu: fluocortolone; Tx: transplantation; P: progesterone; C: control group; ↓: decrease; ↑: increase; ↔: no statistical difference.

**Table 4 biology-10-00271-t004:** Demographical and medical data of cardiac transplant recipients.

Conditioning (Number of Patients)	Sex (Number of Patients)	Age (Years)	Time after Tx	Hormonal Changes	Post-Tx Complications	Literature
A Rapa + MMF/Rapa + Tac (66)	M (132)	A 49.8	A mean 2.6 years	T: ↓ A vs. B LH: ↑ A vs. B FSH: ↑ A vs. B T/LH ratio: ↓ A vs. B	lack of information	[42]
B Tac + MMF (47) CsA + MMF (8) Tac + AZA (4) CsA + AZA (5) Tac (2)		B 49.7	B mean 2.9 years			
A Rapa + MMF (23)	M (132)	49.75	mean 2.7	T: ↔ A vs. B ↓ A&B vs. C LH: ↑ A vs. B ↑ A&B vs. C FSH: ↑ A vs. B ↑ A&B vs. C T/LH ratio: ↓ A vs. B; ↓ A&B vs. C	lack of information	[42]
B Rapa + Tac (43)						
C Tac + MMF (47) CsA + MMF (8) Tac + AZA (4) CsA + AZA (5) Tac (2)						
CNIs + PRE	M (108)	18–70	A: 1 month B: 6 months C: 24 months	A T: 257 ng/dL free T: 6.2 ng/dL LH: 4.5 mIU/mL FSH: 4.2 mIU/mL E2: 27 pg/mL Free E2: 2%	increased BMI (B&C vs. A)	[41]
				B T: 378 ng/dL ^↑^ vs. ^A^ Free T: 9.3 ng/dL ^↑^ vs. ^A^ LH:6.3 mIU/mL ^↑^ vs. ^A^ FSH: 7.3 mIU/mL ^↑^ vs. ^A^ E2: 38 pg/mL ^↑^ vs. ^A^ Free E2: 1.8% ^↓^ vs. ^A^		
				C T: 383 ng/dL ^↑^ vs. ^A^ Free T: 9.4 ng/dL ^↑^ vs. ^A^ LH: 5.2 mIU/mL ^↔^ vs. ^A^ FSH: 5.2 mIU/mL ^↔^ vs. ^A^ E2: 37 pg/mL ^↑^ vs. ^A^ Free E2: 1.7% ^↓ vs^		

M: male; F: female; CsA: cyclosporine A; AZA: azathioprine; PRE: prednisone; Tac: tacrolimus; MMF: mycophenolate mofetil; Rapa: rapamycin; CNIs: calcineurin inhibitors; Tx: transplantation; P: progesterone; C: control group; ↓: decrease; ↑: increase; ↔: no statistical difference.

## Data Availability

Not applicable.

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
