# Peer review of "Hormonal (Im)Balance and Reproductive System’s Disorders in Transplant Recipients—A Review"

_biology, 2021, doi:10.3390/biology10040271_

Round 1

Reviewer 1 Report

In this review, the authors have a performed a very informative summary of the current literature on the topic of organ transplantation (focusing on kidney, heart, liver and bone marrow) and related immunosuppressive therapies and their effect on the reproductive system and hormonal homeostasis.  The review is nicely written, exhaustive and it raises an important aspect of the hormonal imbalance, often neglected in non-purely endocrine/hormonal conditions/diseases.  The reviewer feels that there are though some points which would need to be addressed to make the understanding and the global message easier and more clear for the reader

Major comments:

  1. General: there are too many abbreviations in the text, which make really difficult to follow the text after the first page. I would recommend keeping only the most commonly used abbreviations (e.g. CKD, hormones such as FSH, LH etc.) but spell the other which are not so common to non-clinical readers (ID, IT, RTR, KT etc.).
  2. The renal transplant recipient paragraph is very long and hard to follow. Making it more concise and focused on the main result of each study would help the reader to get the main message of the paragraph. Additionally, a summarizing table for the kidney transplant paragraph where each study and the main finding are reported would also give a nice overview.
  3. This review feels that it would help to have a final table summary where the effect of the analyzed drugs in every organ transplant is represented in relation to hormone level and/or reproductive system

Author Response

Dear Reviewer #1:

Thank you very much for your revision. We really appreciate all your comments and constructive criticism. We have considered all your suggestions, and we have improved the manuscript using red colour; we hope that these changes meet your approval.

Reviewer #1:General: there are too many abbreviations in the text, which make really difficult to follow the text after the first page. I would recommend keeping only the most commonly used abbreviations (e.g. CKD, hormones such as FSH, LH etc.) but spell the other which are not so common to non-clinical readers (ID, IT, RTR, KT etc.).

(response)
We are very grateful for your criticism and suggestion. Following your advice we have reduced the number of abbreviation, leaving only these that are most commonly used and understandable to non-clinical readers.

The renal transplant recipient paragraph is very long and hard to follow. Making it more concise and focused on the main result of each study would help the reader to get the main message of the paragraph. Additionally, a summarizing table for the kidney transplant paragraph where each study and the main finding are reported would also give a nice overview.

(response)

We really appreciate your suggestions and advice, thank you for that. We have shortened the paragraph concerning renal-transplant recipients and also we have reorganized and rephrased it, to make it more clear and accessible to readers. Moreover, following your advice, we have added a summarizing table after each section of our review. We hope that this new version will meet your expectations.

This review feels that it would help to have a final table summary where the effect of the analyzed drugs in every organ transplant is represented in relation to hormone level and/or reproductive system

(response)

Thank you, we are very grateful for this constructive comment. We have added a summarizing table including the treatment therapy, sex, age, number of patients and side effects of the reproductive system, as well as time after transplantation, at the end of each section of our review. Moreover, we have prepared a graphical summarization (Figure 1) of the most commonly noticed hormonal changes based on the transplanted organ. We hope that the current version of our manuscript will meet your approval.

Reviewer 2 Report

Hormonal imbalance (HI) or the alteration of hormonal equilibrium, is a serious health concern for both females and males. Here, the authors summarize the impact of tissue transplantation (Tx) on the imbalance of sex-hormones and their impact on reproduction. They describe several Tx models including Kidney, liver, heart and bone marrow, all of which are essential for survival. The authors also considered the influence of Immuno-suppressive therapies (IT) which are commonly used to prevent allograft rejection following Tx. Overall, the review is intriguing. 

The present review displays a different way of evaluating the connection between Tx, Hormone imbalance and the reproductive system in both males and females. This contrasts with previous reviews (PMID: 31783875, PMID: 30622369) which are gender/Tx model limited or non-sexual hormone related.

The review addresses important questions on hormonal imbalance related to organ/tissue transplantation. Several several can significantly improve it and ease the reading process:

  • The manuscript has considerable writing issues that need to be addressed.
  • Provide a graph summarizing hormonal alterations based on major sites of transplantation.
  • Provide a graphical summary based on gender, hormonal alterations, and recovery time post treatment.
  • Supplementary figure 1: provide more details in your figure legend and describe all acronyms.

The conclusions have enough supportive evidence. However, the authors need to work on reorganizing their data/table/graphs (attached as supplementary) to make their points more obvious to the reader. 

Author Response

Dear Reviewer #2:

We are very grateful for your advice and constructive comments concerning our manuscript. We really appreciate all your comments and constructive criticism. We have considered your suggestions, and we have improved the manuscript. We hope that these changes meet your approval.

Reviewer #2:The manuscript has considerable writing issues that need to be addressed.

(response)We are very grateful for your advice. We have made the language proofreading of the whole the current paper by professional.

Provide a graph summarizing hormonal alterations based on major sites of transplantation.

(response) Thank you for this advice. We have prepared a graph summarizing (Figure 1) the most commonly noticed hormonal changes based on the transplanted organ. We hope that the current version of our manuscript fulfils your expectation.

Provide a graphical summary based on gender, hormonal alterations, and recovery time post-treatment.

(response) We are grateful for this advice, however, with all the respect we are not able to provide such a graph. In the current review, the data is numerous and varied greatly. Moreover, the units for hormone concentration are also different for each hormone. Thus, taking it all into account make it impossible to provide one graphical summary. Nevertheless, we have prepared short summarization in a form of a table at the end of each section of our manuscript and additionally, we have provided a graph summarizing the most commonly observed hormonal changes (increase or decrease) based on the site of transplantation. We hope that the current version meets your approval.

Supplementary figure 1: provide more details in your figure legend and describe all acronyms.

(response) Thank you very much for these important comments. We have provided more details and all the acronyms are now described.

The conclusions have enough supportive evidence. However, the authors need to work on reorganizing their data/table/graphs (attached as supplementary) to make their points more obvious to the reader. 

(response) Thank you for this comment too, we really approve of it. We have reorganized our tables and graphs to make our review more accessible and clear for readers.

Round 2

Reviewer 1 Report

The authors have answered all raised points and the reviewer believe that the manuscript in the current status is suitable for publication.